# CTD Sensors for Ocean Investigation Including State of Art and Commercially Available

**DOI:** 10.3390/s23020586

**Published:** 2023-01-04

**Authors:** Shiyu Xiao, Mingliang Zhang, Changhua Liu, Chongwen Jiang, Xiaodong Wang, Fuhua Yang

**Affiliations:** 1Engineering Research Center for Semiconductor Integrated Technology, Institute of Semiconductors, Chinese Academy of Sciences, Beijing 100083, China; 2College of Materials Science and Opto-Electronic Technology, University of Chinese Academy of Sciences, Beijing 100049, China; 3Institute of Oceanology, Chinese Academy of Sciences, Qingdao 266031, China; 4National Laboratory for Computational Fluid Dynamics, School of Aeronautic Science and Engineering, Beihang University, Beijing 100191, China; 5Beijing Academy of Quantum Information Science, Beijing 100193, China; 6Beijing Engineering Research Center of Semiconductor Micro-Nano Integrated Technology, Beijing 100083, China

**Keywords:** CTD, electrical conductivity, temperature, depth, pressure sensor, ocean sensor

## Abstract

Over 70% of the earth’s surface is covered by oceans; globally, oceans provides a huge source of wealth to humans. In the literature, several sensors have been developed to investigate oceans. Electrical conductivity temperature depth (CTD) sensors were used frequently and extensively. Long-term accurate CTD data is important for the study and utilization of oceans, e.g., for weather forecasting, ecological evolution, fishery, and shipping. Several kinds of CTD sensors based on electrics, optical, acoustic wave and radio waves have been developed. CTD sensors are often utilized by measuring electrical signals. The latest progress of CTD sensors will be presented in order of performance. The principles, structure, materials and properties of many CTD sensors were discussed in detail. The commercially available CTD sensors were involved and their respective performances were compared. Some possible development directions of CTD sensors for ocean investigation are proposed.

## 1. Introduction

There is a huge amount of information hidden in the ocean which is critical for humanity to make use of ocean resources. The information consists of the properties of the marine body, ocean acoustic environment or marine geology and geomorphology. Thus, to obtain this information, many ocean observation techniques have been developed. Modern ocean observation techniques are multidimensional and can be divided into satellite-based observation, on-ship observation and underwater observation [1]. Satellite-based observation monitors the ocean by remote sensing techniques and has the advantage of utilizing a wide observation area and monitoring over a long period of time [2]. On-ship observation is the earliest and most commonly used method in ocean observation. There are fewer restrictions on size, power supply and memory capacity of sensors. With ships carrying measurement equipment, we can obtain precise in situ information. Underwater observation benefits from the development of wireless transmission and underwater glider techniques. Underwater gliders facilitate deploying autonomous gliders into harsh marine environments with a relatively low cost compared to traditional ship-based observation [3,4,5]. All the presented observation methods rely on various sensors, such as CTD (conductivity, temperature and depth) sensors, DO (dissolved oxygen) sensors and pH sensors. 

Sensors are tools we use to discover information contained in the ocean. Conductivity, temperature and depth (CTD) sensors are especially important among all those sensors measuring parameters of seawater. They provide critical physical properties (conductivity, temperature and depth) of oceans which are parameters of sea water equation of state and essential for ocean dynamics modelling [6,7,8,9]. In addition, these three parameters provide necessary parameters for other marine sensors and models of ocean environment, such as sound velocity in seawater [10,11]. CTD data is widely used for the investigation of climate change in oceans [12,13,14,15,16], seabed environment surveys [17,18,19,20], local hydrologic properties [21,22,23,24,25,26,27,28,29,30,31,32,33] and predictions of ocean phenomena [34]. In 2008, M. F. Santini analyzed water masses and sea ice formation rates using data from CTD sensors attached to southern elephant seals [35]. In 2014, R. Sorgente constructed a modelled sea current based on CTD data to forecast drifter trajectories [36]. Researchers in Japan using CTD data revealed short-term excursions of Japanese sea bass from sea water to fresh water [37]. To collect temperature and depth information of the upper ocean quickly, accurately and widely, the Argo program is proposed. Argo [38] is a part of the GOOS (Global Ocean Observing System) [39]. The program deploys floats carrying CTD sensors in the global ocean and until now, over 3000 floats have been deployed [40]. Argo continuously provides data of the upper 2000 m of the ocean which contributes to observe, understand and forecast ocean variability. So far, it has provided over 2,000,000 profiles for ocean research [41]. To achieve long-term and accurate observation, stable and sensitive CTD sensors with high precision are required.

The typical CTD sensors are electronic systems with high precision and wide practicality. So far, many commercial CTD sensors developed for high-precision measurement of these parameters use electrical methods. For example, SEB 911 plus uses conductivity cells for salinity measurement, NTC thermistors for temperature measurement and quartz crystal resonators for depth measurement. However, most commercial CTDs are expensive, large in size and susceptible to electromagnetic interference. To solve these problems, recently, MEMS-based CTD sensors and fiber sensors have been studied [42]. Although they are not as stable and accurate as commercial sensors, these studies exhibit wide application potential and future development directions of CTD sensors. Based on introduction of CTD sensors of different sensing principles, this article discusses conductivity, temperature and depth sensors, respectively and gives a comprehensive comparison of commercial sensors. The working principles, structures of sensors and comprehensive properties are analyzed to provide a potential developing direction of CTD sensors.

## 2. Conductivity/Salinity Sensors

Salinity is of great importance in calculating and modelling oceanographic variables. Many phenomena and processes occur in the ocean, often related to the distribution and variation in salinity [43]. Absolute salinity (S_A_) is defined to be a scale of total mass of substances dissolved in seawater [44]. However, S_A_ often refers to ‘Density Salinity’ or S_A_^dens^ due to the difficulty of measurement. The measurement of density is a good way to estimate absolute salinity because density takes into account all mass constituents in seawater. Optical measuring techniques can be a good way to measure absolute salinity [45]. The refractive index of seawater is related to the density of seawater. The relationship between refractive index and density is given by the Lorentz-Lorenz equation:(1)$$\frac{n^{2}-1}{n^{2}+2}=\frac{m_{r}\rho }{W}$$
where n is the refractive index of seawater and m_r_ is the molar refractivity, ρ is the density of seawater and W is the molecular weight of the species present in fluid. However, there is another concept of salinity, namely ‘practical salinity’ which is defined to use the conductivity of seawater to estimate the salinity of seawater. Most measurement results of electrical CTD sensors actually use the definition of practical salinity [46,47]. The typical accuracy for marine environment monitoring is about 0.1 psu, which means the accuracy of the conductivity sensor is higher than 0.1 mS/cm [48]. The performance of electrical conductivity sensors can achieve an accuracy of ±0.0003 S/m, a resolution of 0.00004 S/m and a stability of 0.0003 S/m per month in the range of 0–7 S/m. Mainly, there are two types of electric conductivity sensor for ocean salinity monitoring: electrode conductivity sensors and the inductive conductivity sensors. Inductive sensors measure conductivity by electro-magnetic induction. They can avoid polarization effect and work near the surface since there is no electrode in the sensor [49]. However, the size of an inductive sensor is large because of a toroidal transformer which limits its miniaturization. Electrode sensors measure salinity with a conductivity cell whose parameter corresponds to the size and shape of electrodes [50]. The design of a conductivity cell is the most important aspect of electrode conductivity sensor design. Common electrode sensors can be divided into two-electrode type, four-electrode type and seven-electrode type, according to number of electrodes. Four-electrode sensors and two-electrode sensors are used in commercial sensors more often. In recent years, optical salinity sensors have been developing rapidly. The refraction index of sea water is closely related to salinity, temperature and depth. This relation can be described by an empirical formula [51,52]: (2)$$\begin{array}{ll} n=1.3247+3.3 & \times 10^{3}\lambda^{-2}-3.2\times 10^{7}\lambda^{-4}-2.5\times 10^{-6}T^{2}\\ & +\left(5-2\times 10^{-2}T\right)\left(4\times 10^{-5}S\right)\\ & +\left(1.45\times 10^{-5}P\right)\left(1.021-6\times 10^{-4}S\right)\left(1-4.5\times 10^{-3}T\right) \end{array}$$
where *n* is the refraction index of seawater, λ is the wave length of incident light, *T* is the temperature of seawater, *S* is the relative salinity of seawater and *P* is the pressure. Unlike traditional electrical CTD sensors, optical sensors do not measure salinity by measurement of conductivity. So, it is more accurate to refer to it as a Salinity-Temperature-Depth (STD) sensor. However, we still use optical CTD sensor to indicate the optical STD sensor in order to follow the customary term CTD. Based on the relationship above, optical CTD sensors are developed. Optical salinity sensors are easy to process and miniaturize. Determined by principle of measurement, it is easy to design dual-parameter or multi-parameter optical sensors. Though they have not been used in practice widely, they have great application prospects. 

### 2.1. Inductive Sensors

Kanghui Song [53] demonstrated a transformer type inductive sensor based on an extended model to analyze the sensor. The sensing principle is shown in Figure 1a. Figure 1b exhibits the equivalent circuit of the sensor. The alternative voltage applied on the drive coil induces an ionic electric current in the solution around the sensor. The current in solution induces a current in sensing coil and Rs in Figure 1b is the equivalent resistance of solution. The current in sensing coil is proportional to conductivity of the solution. The performance of the conductivity sensor is conducted in sulphuric acid solutions which had 1%, 3%, 5%, 8% concentrations and the results are shown in Figure 1c. The sensitivity of the sensor is about 4 mV per mS/cm and the measuring range is up to 350 mS/cm. Stability is measured in 0.15 M sodium hydroxide solution at 25 °C and error in long term measurement is less than 1.8 mV. The sensor has a response time within 1 s and the error over time is less than 1.8 mV. Measuring error is obtained by comparing with commercial sensor Water Quality Meter F-74 at temperature ranges from 15 °C to 45 °C. As shown in Figure 1d, The maximum error is 0.6mS/cm which appears at 20 °C. An integrated and autonomous CTD SZQ1-1 applied for underwater glider is developed by Bin Lv from Qilu University of Technology [54]. An inductive conductivity sensor is integrated in this sensor. In depth range of 0–100 m, SZQ1-1 shows an average error of −0.225 PSU compared with SBE 19plus. However, in range of 100–500 m the error expanded. The error is caused by thermal lag effects, different velocity of underwater glider and this sensor has room for improvement.

### 2.2. Electrode Sensors

A direct-reading MEMS conductivity sensor with a parallel-symmetric four-electrode configuration which integrates a silicon-based platinum thin film strip electrode and a serpentine temperature compensation electrode was proposed by Zhiwei Liao [50]. As shown in Figure 2a, the voltages and current electrodes are separated to weaken the polarization phenomenon. COMSOL FEM simulation is used to determine the dimension of the electrodes which is closely related to parameter of electrode cell. The equivalent circuit is shown in Figure 2b. When an excitation signal is applied to the circuit, two voltage electrodes measure the voltage drop across the equivalent resistance Rw2 of solution and two current electrodes measures the current across Rw2. So, the conductivity of solution can be obtained by current and voltage measured by electrodes. The maximum range of the sensor is 107.41 mS/cm. Between 0 and 76.422 mS/cm, the measuring precising ranges from 0.005 mS/cm to 0.165 mS/cm. Between 81.879 mS/cm and 107.41 mS/cm, the precising ranges from 0.229 mS/cm to 0.401 mS/cm. The standard deviation is shown in Figure 2c. The maximum measuring error was 0.073 mS/cm at 57.6772 mS/cm which is illustrated in Figure 2d. The sensor provides an interface for real-time reading of conductivity value which is significant for on-site observation. Recently, Chaonan Wu [55] introduced a four-electrode high precision conductivity and temperature sensor for ocean measurement and its batch microfabrication. The batch fabrication process is shown in Figure 3. This processing is manageable for batch fabrication and materials used are cheap enough for large-scale application. The study of batch fabrication is significant for commercial use. A good manufacture processing can greatly reduce the cost of production, and reducing the cost of producing can promote the application of new developed techniques which is a benefit for researchers. In addition, progress in the batch fabrication of CTD sensors can decrease the cost of building a CTD sensor array for large-scale ocean collaborative observation. The chip size is approximately 12 mm × 12 mm and a four-inch silicon substrate contains 34 chips. The performance of the sensor for a batch is considered good with a consistency of 0.0048 mS.cm and an accuracy of 0.08 mS/cm. Xi Huang [48] introduced a low-power, miniature seven-electrode type CT sensor. Seven-electrode type sensors can avoid proximity effects. The working principle is schemed in Figure 4a. Electrodes 1–4 work the same way as in four-electrode type sensors. Electrode 7 is used to form an axially symmetric electric field by applying the same voltage as electrode 1 and electrode 5, 6 which is not connected to any circuit keep the cell axially symmetric. As shown in Figure 4b, when the conductivity of solution ranges from 25 to 55 mS/cm, the error of the sensor is distributed over the range 0.03 mS/cm. The CT sensor system is tested on an Atlantic cruise in 2010. The system which has a battery life of one month were packaged in 10 × 15 cm pressure pots, and deployed from a floating buoy. However, experiments identified some weaknesses with long-term stability which may be caused by water uptake of the epoxy insulators. Boron doped diamond electrodes can be used in conductivity sensors. Boron doped diamond electrodes exhibits great response time and high long-term stability [56]. Xu Mingxia introduced a salinity sensor with boron doped diamond electrodes. Long-term error detection of this sensor can reach 0.005 mS/cm [57].

### 2.3. Optical Fiber Sensors

A hybrid fiber device consisted of a hollow-core fiber (HCF)-based Fabry-Perot interferometer (FPI) and no-core fiber (NCF)-based anti-resonance (AR) structure was demonstrated to measure seawater temperature and salinity [58]. The sensing principles were schemed in Figure 5a. An FPI part was used to measure salinity and NCF-AR structure worked for temperature detection. The measuring system was shown in Figure 5b, where seawater was pumping in and out and the sensing capsule was kept in a thermostat. The reflection signals from FPI were used to detect salinity, shown in Figure 5c. Under 24 °C, the standard seawater solutions with concentrations of 0‰, 5‰, 20‰, 30‰, 35‰, 40‰ were investigated with sensitivity of the FPI spectra shifts to a long-wavelength direction as a function of salinity increases with a slope of 0.235 nm/‰. However, the maximum relative error was 5.48%, shown in Table 1. A dual parameters salinity and pressure sensor based on surface plasmon resonance is developed by Musen Yang [59]. Salinity is measured by resonance between evanescent wave in single-mode fiber and surface plasmon wave on gold film around the single-mode fiber. The sensor has a salinity sensitivity of 0.36 nm/‰ and a measurement range of at least 0–60‰. 

### 2.4. Acoustic Sensors

With the increasing accuracy of sound velocity measurement, it is possible to use sound velocity to estimate absolute salinity of seawater. Sound velocity is a thermodynamic quantity directly linked to adiabatic compressibility of the medium [45]:(3)$$c=c\left(S_{A},t,p\right)={\left(\partial P/\partial \rho \right)}^{0.5}={\left(\rho k\right)}^{0.5}$$
where k is the adiabatic compressibility coefficient. K is expressed in Pa^−1^. c is the speed velocity, S_A_ is the absolute salinity of seawater, t is the temperature of seawater, p is the pressure and ρ is the density, c is inversely proportional to the density of seawater and thus directly related to absolute salinity.

The Russian Academy of Science developed a module for calculating salinity using temperature and sound velocity—SVT module [60]. The deviation range of the results of the sound velocity measurement was ±0.02 m/s. The error of temperature measurement was below ±0.004 °C. The deviation range of salinity calculating was ±0.03‰. However, to meet oceanographic requirement, we still need to conduct research on more accurate sound velocimeter. Additionally, diffraction effects and compensation limits practical performance of acoustic salinity sensor.

### 2.5. Radio Sensors

Radiometers are used for large-scale sea surface salinity (SSS) monitoring. In 1990, SINGH proposed a way to estimate sea surface salinity using Klein-Swift model by measuring the brightness temperature over sea water [61]. Until now, we still use inversion of K-S model to estimate sea surface salinity by the brightness temperature over sea water. Usually we use the L-band radiation (around 1.4 GHZ) to obtain the brightness temperature of sea surface [62]. SMOS (Soil Moisture and Ocean Salinity) is one of ESA’s Earth Explorer missions [63]. It carries an L-band novel interferometer radiometer. The SMOS mission has been providing sea surface data since 2010. Radio sensors make up for the large-scale monitoring capability that in situ sensors do not have, which is an important part of ocean observation networks.

## 3. Temperature Sensors

The temperature range of the ocean is approximately −5~35 °C [64], which is a relatively narrow and easy to achieve measurement range for most temperature sensors. However, a high precision of 0.02 °C for temperature measurement is required [65]. Temperature from −5 to 35 °C could be measured with resolution of 0.0002 °C, stability of 0.0002 °C per month and accuracy of ±0.001 °C. Temperature sensors used for ocean monitoring can be divided into typical electric sensors and optical sensors. So far, the most widely used electric temperature sensors in CTD sensors are platinum resistance temperature sensors. Platinum is useful because of its stability [66], antioxidation, linear response to temperature [67] and high purity. With a high melting point of 1772 °C [68], platinum resistance can be easily used in most cases. When calibrated properly, a platinum resistance sensor can achieve accuracy of less than a 0.001 °C [69]. In addition, compared with other temperature sensors, platinum resistive sensors are smaller which means a faster response time [70]. Thermistors are also commonly used for temperature measurements. The temperature sensor of SBE-911 is a thermistor. Thermistors are resistors which are sensitive to temperature and usually have a negative temperature coefficient. Within the operating temperature range, the resistance of a thermistor can vary by several orders of magnitude. Thermistors are generally ceramic or made of a polymer. A ceramic thermistor also exhibits a good stability for long-term measurement. Compared with platinum resistors, thermistors have high accuracy but limited operating range [71].

### 3.1. Optical Fiber Sensors

Optical sensors have been developing rapidly in recent years for their size, cost, networking and multi-parameter sensing but with disadvantages of low mechanical strength [51]. In Figure 5a, the reflective index of polymer coating [72,73,74] on NCF-AR structure was temperature-sensitive. The seawater salinity is kept at 0‰ and the temperature increases from 15 °C to 35 °C, the transmission signals of the NCF-AR structure have a slope of −4.628 nm/°C in the range of 15 °C to 25 °C. In the range from 25 °C to 55 °C, a larger sensitivity of −4.948 nm/°C was obtained with another resonant wavelength, shown in Figure 5d. However, the minimum relative error of temperature measuring was 0.76%, showed in Table 1. Sensitivities are −4.948 nm/°C and 0.235 nm/‰ salinity characteristic is tested.

A dual-parameter fiber-optic sensor consisting of a side-polished microcavity in a single mode fiber (SMF) and a polydimethylsiloxane (PDMS) coating was proposed by Musen Yang in 2022 [75]. Figure 6a exhibits the structure of the sensor. Multiple modes excited in the inside coating forms the first Mach-Zehnder interferometer (MZI). The second MZI is formed by light reflecting in the waist region interfering with light in path 1. Two MZIs measure temperature and pressure separately. When the temperature and pressure change, dip shifts on the transmission spectrum of two MZI are observed. As shown in Figure 6b,c, when temperature ranges from 25 °C to 60 °C in steps of 5 °C, the wavelength of dip and peak shows a linear response to temperature with the sensitivity of −1.19 nm/°C and −0.22 nm/°C. An encapsulation for microfiber MZ interferometer temperature and salinity sensor was developed by Li-Hui Zhang [76]. The encapsulation consists of a C-shape stainless steel groove with a slit on the bottom and polymer as adhesive. After encapsulated by modified acrylate adhesive (MAA) or transparent epoxy AB adhesive (TEAB), the sensor shows better tensile withstanding and linearity sensitivity response.

Junyang Lu conducted studies on pressure and temperature dual-parameter sensors based on polymer-coated tapered optical fiber [77,78,79]. This sensor is based on their previous study of an optical microfiber couple combined Sagnac loop [80]. The schematic structure of the sensor is shown in Figure 7a. Light is injected from Port 1 and will be divided into two parts. Those two parts will be reflected by Faraday rotating mirrors (FRMs) in Port 3 and Port 4, and then in the waist region reflection lights couple and interfere with each other. By monitoring the shift in characteristic wavelength of the output light spectrum in Port 2, the temperature and pressure can be measured. As shown in Figure 7b, the interference spectrum is measured when temperature ranges from 21.5 °C to 28 °C. The interference spectrum has a significant blue shift related to the temperature. Figure 7c shows the sensitivity of different sensing dips. The maximum sensitivity is −2.283 nm/°C. Five tests are conducted to evaluate error of the sensor. There is a maximum error of temperature measuring of 2.14%.

### 3.2. Acoustic Sensors

The speed of sound is dominated by water temperature, density and pressure [45]. Typically, the speed of sound near the ocean surface is about 1520 m per second, which is more than four times faster than it in air. The speed of sound in water increases with an increasing water temperature, increasing density and increasing depth. Most of the change in the speed of sound in the surface ocean is due to changes in temperature. To measure the temperature of the water, a sound pulse is sent out from an underwater sound source and recorded by a hydrophone in the water at a defined distance. The time the sound takes to go from the source to hydrophone is measured. From the travel time, the speed of sound between the source and the hydrophone can be calculated. If the density and depth where the sound traveled are known, the temperature of the water can be calculated.

There are two specific methods of measuring the temperature of the ocean with sound. Acoustic Tomography uses precise measurements of acoustic travel times to draw ocean temperature maps. Data from many crossing acoustic paths are used to generate these maps of ocean temperatures [81]. Inverted Echo Sounders (IES) measure the temperature of the water column at a single point. The source emits a sound pulse from the bottom to surface of the ocean. The IES listens to the returning pulse from the ocean surface. The travel time of the sound is used to calculate the speed of sound. The temperature profile is calculated from the speed of sound [82]. IES is often used to monitor a particular region of the ocean with groups (or arrays) to cover a large area.

### 3.3. Radio Sensors

The surface temperature of seawater (SST) has been observed using microwave sensors and thermal infrared sensors since 1970s [83]. In 1981, the Seasat satellite achieved a root-mean-square sensitivity of 1.2 °C of SST measurement via the Seasat multichannel microwave radiometer [84]. Microwave radiometer measures SST by measuring the brightness temperature which is proportional to the SST. Thermal infrared sensors measure the brightness of the black-body radiation which is related to the SST. Images obtained by thermal infrared sensors have a higher resolution because of their shorter wavelength. Microwave radiometer is featured by cloud penetration and all-day-round function [85]. In practice, different bands of electromagnetic wave used for remote sensing are mutually complemented.

## 4. Depth Sensors

The depth of oceans is mainly measured through pressure sensors to obtain seawater pressure and then through the formula calculation. Depth can be calculated by the following formula:(4)$$h=\frac{P}{g\rho }$$
where h is the depth, P is the pressure, g is the acceleration of gravity and ρ is the density of seawater. According to the formula, as depth increases by around 10 m, pressure rises by one atmosphere. The deepest part of the ocean can reach 11,000 m, and commonly CTD sensors work in the upper 2000 m of the ocean. So, pressure sensors used for ocean monitoring have to meet the requirement of a wide range, good stability and accuracy in high pressure. A depth of 10,500 m could be reached with a resolution of 1.0 m and 400 ppm drift per year. High-pressure sensors reported in recent years can be divided into resonant quartz crystal sensors, optical fiber sensors and piezoresistive sensors. Piezoresistive sensors are widely used for high pressure measuring. There has been over 50 years of development of piezoresistive sensors [86]. Their fabrication and calibration are well developed. Batch fabrication and miniaturization of piezoresistive sensors is easy with great precision by silicon micromachining technique [87]. However, the accuracy of the piezoresistive method is relatively low (0.1%FS) [88]. Resonant pressure sensors indirectly measure pressure through inherent frequency change caused by external pressure which leads to deformation. Resonant quartz pressure sensors have high precision but are expensive and complex to fabricate. The accuracy of resonant quartz pressure sensors can achieve 0.021%FS [89]. Optical sensors are also used for pressure measurement, but they are weak in mechanical strength and not used in commercial CTDs yet. In addition, resonate silicon crystal pressure sensors are cheaper than resonate quartz crystal sensors, and they are already commercially available (e.g., Druck and Yokogawa Electric Co., Tokyo, Japan).

### 4.1. Resonate Sensors

An integrated resonate-diaphragm structure sensor is demonstrated for high pressure measurement [88,90]. The sensor consists of a glass wafer and a silicon-on-insulator (SOI) wafer which is integrated with a dual resonators and diaphragm. As shown in Figure 8a, the sensor consists of a SOI wafer which is used to form resonators and diaphragm and a glass wafer used for vacuum package. The resonators are formed in a device layer and coupled with the oxide layer while the pressure sensitive diaphragm is on the handle layer and has the same thickness as the handle layer. When a high-pressure is applied on the diaphragm, the deformation of the diaphragm causes changes in stresses in the diaphragm which then transmit to the resonators. Caused by stresses, the frequency of resonators I increase and frequency of resonator II decreases. The differential outputs of dual resonators can decrease temperature sensitivity of sensor. The pressure sensitivities of the sensors are 0.205 kHz/MPa and −0.211 kHz/MPa which are studied by finite element analysis and are schemed in Figure 8b. The two resonators with similar material properties can reduce temperature drift by differential outputs. Temperature sensitivity is tested by loading temperature under static pressure. The results are shown in Figure 8c, sensitivity is reduced from 50.5 Hz/°C to 0.9 Hz/°C. Measurement errors of the sensor are within 800 Pa in temperature range of −10 °C to 60 °C and pressure range of 0 MPa to 5 MPa, shown in Figure 8d. The accuracy of this type of sensor is better than 0.2% in the pressure range of 110–6500 kPa and the temperature range of −10 °C–60 °C. This sensor exhibits a good performance in laboratory testing. However, it still needs to be tested in a sea trial. Only limited pressure is achieved in a laboratory environment and measurement scale and long-term stability are key factors which also need to be considered. A high sensitivity quartz resonate pressure sensor based on a quartz double-ended tuning fork (DETF) was proposed by Quanwei Zhang [91]. In this sensor, a quartz tuning fork is used as a pressure sensitive component which is insensitive to external environmental factors compared with other resonate pressure sensors. The sensitivity of the sensor achieves 36.58 Hz/kPa. The measurement range in a laboratory is tested to be less than 100 kPa. The operating temperature of this is in the range of −20 °C to 60 °C which is suitable for seawater measurements.

### 4.2. Piezoresistive Sensors

Ting Li introduced a piezoresistive pressure sensor with high accuracy and high sensitivity [92]. The piezoresistive sensor is designed based on piezoresistive phenomenon of silicon that the resistance changed when a pressure is applied on silicon. The sensor consists of a thin diaphragm and a Wheatstone bridge structure. When the diaphragm is subjected to external pressure, the resistance of the diaphragm changes and the changes are measured by Wheatstone bridge structure. The structure of the sensor is shown in Figure 9. The pressure sensor is tested under pressure at 0 MPa, 24 MPa, 48 MPa, 72 MPa, 96 MPa, and 120 MPa, shown in Table 2. The related average values of the piezoresistive sensor parameters are shown in Table 3. The pressure sensor has a pressure sensitivity of 0.425 mV/MPa, accuracy of 0.0182% and pressure range of 0–120 MPa. Another piezoresistive pressure sensor was designed and fabricated by Mengru Jiao [93]. An optimized bulk-micromachining technology and a package using Si-glass bonding technology is developed to fabricate this sensor. The sensor has a pressure sensitivity of 33.04 mV/MPa over a wide pressure range of 0–5 MPa and a maximum error of 0.43% with a chip size of only 1.5 mm × 1.5 mm × 0.82 mm. A temperature compensation system consisted of a temperature sensor and a microcontroller unit is developed to overcome temperature drift of the sensor. With this compensation system, the output pressure becomes weakly correlated with temperature.

### 4.3. Optical Sensors

In Figure 5a, different optical paths can form different sensing phenomena. The MZI structure can be also pressure sensitive. The pressure sensitivity of the sensor is −3.96 nm/MPa which is tested in a range of 0 MPa to 0.48 MPa.

As shown in Figure 7a, the microfiber structure is pressure sensitive. Pressure sensitivity is obtained by measuring interference spectrum in pressure between 0 MPa to 3 MPa. The result is shown in Figure 7b,c, with the increasing pressure there is a red shift of dips in interference spectrum. The pressure sensitivity of sensor of dip1, dip2, and dip3 is 2.63 nm/MPa, 2.927 nm/MPa and 3.301 nm/MPa. The maximum error is 11.25% at 4 MPa. The repeatability of pressure measurement of the sensor is tested by five increase and decrease tests under the same conditions. The average error of the sensor is less than 2.9%. A high precision fiber optic Fabry-Perot pressure sensor with pressure sensitivity of 257.79 nm/MPa is introduced by Yanan Zhang [70]. This pressure sensor is based on an AB epoxy adhesive film. The AB epoxy adhesive film forms one side of FP cavity and deforms under pressure. However, the measurement range is relatively small. This sensor can only measure pressure up to 70 kPa which means it can only work in the surface of the ocean.

## 5. Commercially Available CTD Sensors

Different from sensors in labs, commercial CTD products have a complete calibration system and a corresponding compensation algorithm. A calibration system can greatly affect the accuracy of long-term measurement of CTD sensors. It is aimed to decrease the influence of error accumulation. Errors caused by a lack of calibration cannot be corrected later by the algorithm. Different commercial CTD sensors have their unique calibration methods and new calibration methods are continuously being developed by researchers [94,95,96,97]. Data correction algorithms are another factor affecting the authenticity of CTD data. Measurement results of individual sensors are always affected by environments and the method of measurement, such as thermal lag, effects of temperature measurement, temperature drift of pressure sensor [98,99,100,101,102,103,104,105], etc. These errors can be partly corrected via compensation algorithm [106,107,108,109,110,111]. For example, temperature drift of piezoresistive pressure sensors can be corrected by a temperature compensation algorithm. An algorithm developed by S.Wu can realize 0.03% accuracy in the range of −6~50 °C and a full scale 60 MPa range [112]. The results of the compensation algorithm are shown in Figure 10. The compensation helps the pressure sensor insensitive to temperature. After compensation, the slope of the output voltage-temperature curve is nearly zero, which mean the error caused by temperature varying is effectively corrected.

The representative companies and their CTDs were shown in Figure 11. Fifteen kinds of CTDs with their respective performances were summarized in Table 4. In addition, the data in Table 4 shows the typical values claimed by manufacturers. The highest accuracy conductivity and temperature were 0.0015 mS/cm and 0.0015 °C, provided by Idronaut-OS333. The highest accuracy depth was ±0.015%FS and the range could reach 10,500 m. Several other properties, such as stability, response time, sampling, power, housing material, size, and weight were also important for CTD applications. 

The conductivity cell of FSI-ICTD (Figure 11a) employed an inductive cell with long-term calibration stability, while its free flushing design avoided thermal contamination and the need for pumps. The silicon pressure sensor provided a superior performance and reliability for depth. A standard-grade platinum resistance thermometer for primary temperature measurement with the world standard (ITS-90 temperature measurement) was utilized. The stability of conductivity, temperature and depth were ±0.0005 mS/cm/month, ±0.0002 °C/month and ±0.002%FS/month. The platinum thermistor had a response time of 150 m/s. The CTD had the highest sampling rate of 32 Hz. 

The Idronaut-OS333 (Figure 11b) had a high-accuracy seven platinum ring quartz conductivity cell with an 8 mm diameter and 46 mm length quartz tube, which guaranteed self-flushing and minimal fouling. An optional UV LED with 280 nm lights was integrated into the conductivity cell, which could sterilize the sample under measurement and avoid the early growth of biofouling inside the quartz cell tube. Temperature was obtained from a very fast response platinum resistance thermometer with negligible self-heating effect. The depth was measured by a high precision pressure transducer based on a piezoresistive bridge, floating on oil, with drift-free sensor interface. Temperature dependency and non-linearity of pressure sensor were mathematically compensated by the interfacing electronics. Titanium housing with 75 mm diameter and 630 mm length confirmed usage at 7000 m depth. 

The RBR-argo^3^ CTD (Figure 11c) was designed for Argo buoys with a compact titanium housing rated at 6000 m depth. The low-power design consumed only 20% of the energy of similar products. The conductivity cell was unaffected by surfactants and was not damaged by drying out. Accurate conductivity to within 10 cm of air-ocean interface could be obtained. Atmospheric measurements provided helpful drift references for interior calibration. The depth was determined by piezoresistive pressure sensors, which was protected by a clear plastic guard. A thermistor type temperature sensor with digital compensate could provide precise measurement. 

Sea Sun Technology company provided 115M CTD, shown in Figure 11d. The seven-pole-cell was used in conductivity sensor. Temperature was obtained by Pt100 with four-pole configuration. The pressure transducer had a piezoresistive full bridge with a diameter of 15 mm and a total height of 6 mm. The casing and diaphragm were made of Hasteloy. The transducer included a temperature compensation of the pressure measurement. The CTD with titanium housing could worked at 500 m depth, and its response time was 150 ms. 

The SBE 911plus CTD was a representative product of SeaBird Scientific Company, shown in Figure 11e. Its unique internal field conductivity cell with exclusive use tube could minimize salinity spiking. The SBE 4 conductivity sensor is a modular, self-contained instrument with a measurement range of 0 to 7 S/m which covers the full range of lake and oceanic applications. Noise and corrosion will be eliminated by an electrically-isolated power circuit and optically coupled output. Cylindrical flow through borosilicate glass cell with three platinum electrodes inside and the arrangements exhibited advantages over inductive or open external fields cells. The electric field is confined inside the cell since the outer electrodes were connected, making the measured resistance independent of calibration bath size or proximity to protective cages or other objects. The output frequency of a Wien Bridge oscillator circuit is controlled by the cell resistance. 

SBE 3 plus is adopted to be the temperature sensor of 911plus CTD, which was calibrated to ITS90 temperature using Sea-Bird’s computer-controlled calibration baths. The baths which are extremely insulated provide a uniform toroidal circulation, yielding an overall transfer accuracy against an SPRT (sequential probability ratio test) within 0.0002 °C. The repeatability of each twelve individually mapped sensors positions was better than 0.0001 °C. The calibration baths were underpinned by the Sea-Bird’s metrology lab. The temperature precision of 50 µK and accuracy of 0.0005 °C are achieved by following consultation with the U.S. National Institute of Standards and Technology. A glass-coated thermistor bead was protected in 0.8 mm diameter thin-walled stainless steel tube. The thermistor, which is exponentially related to temperature, was the controlling element in an optimized Wien Bridge oscillator circuit. The resulting sensor frequency was inversely proportional to the square root of the thermistor resistance and ranged from approximately 2 to 6 kHz, corresponding to −5 to +35 °C. Built-in acquisition circuits and frequency outputs offered the advantage of calibration as separate modules.

The accuracy of the overall system is limited only by the accuracy of the CTD’s master clock, after being individually calibrated in Sea-Bird’s computer-controlled super-low-gradient calibration baths. In the SBE 911plus, clock error contribution was 0.00016 °C, based on a five-year worst case error budget, including ambient temperature influence of 1 ppm total over −20 to +70 °C, plus 1 ppm first year drift, plus four additional year’s drift at 0.3 ppm/year, so errors from this source were demonstrably negligible.

An aged and long history of exceptional accuracy and stability of Digiquartz pressure sensors with temperature compensation was used in 911plus CTD. The Digiquartz pressure sensor also provided a variable frequency output. There is a semiconductor sensor embedded in the pressure sensor to compensate the small ambient temperature sensitivity of the Digiquartz. The sensor frequencies were measured using high-speed parallel counters and the resulting digital data were transmitted serially to the Deck Unit in the form of count totals. The Deck Unit then reconvert the digital data to numeric representations of the original pressures. The superior performance of the Digiquartz instruments was achieved by using a precise quartz crystal resonator whose frequency of oscillation varied with stress induced by pressure. Quartz crystals were chosen for the sensing elements because of their remarkable repeatability, low hysteresis, and excellent stability. An oscillator which is similar to those used in precision clocks and counters is used to maintain and detect the resonate frequency output. The resolution of the pressure sensor is up to one part-per-billion of the full scale. Even under hard environmental conditions, the typical accuracy of 0.01% of the full-scale is still achieved. Other desirable characteristics including high reliability, low power consumption, and excellent long-term stability were still available. In addition, with titanium housing, the depth of 10,500 m could be reached. 

In Aanderaa-SeaGuard CTD, the conductivity sensor 4419R was based on an inductive principle. This facilitated stable measurement without electrodes which were easily fouled and wear out in the field. The temperature sensor was based on a thermistor-bridge. A digital signal processor controlled the sampling of the bridge and calculated the calibrated temperature in engineering units. The pressure sensor was based on a silicon piezo-resistive bridge and temperature drift was compensated by an advanced digital signal processor. 

The NBOSI conductivity sensor was based on a novel four-electrode conductivity cell with an internal electric field (US Patent 6720773). The NBOSI CT sensor consisted of a “fin” style, internal field, four-electrode conductivity cell with an integral, pressure-protected thermistor and a high precision, self-referencing electronics board. 

OS200 CTD was manufactured by Ocean Sensors Company. The conductivity sensor was a conduction type sensor consisting of four electrodes, with electrode pairs located on opposite sides of the sensor substrate. The small cylinder bonded over the electrodes reduces the volume over which the measurement was taken. This sensor design provided repeatability of measurements when subjected to oceanic pressures and temperatures. The response of the conductivity cell depended on the flow of water through the sensor. The conductivity sensor had an open design, providing good flushing characteristics and a high time response. The temperature sensor was a thermistor that changes its resistance with the temperature; it was located at the base of the CT sensor just above the black portion of the sensor. Heat diffusion occurs over time at the surface of the sensor. Therefore, the temperature sensor response was best described as a time averaging of the signal and not a spatial averaging. The sensor was very small and delicate. It should be protected from impact and curious fingers. The pressure sensor was a resistive strain gauge bridge diffused into a silicon diaphragm. The pressure sensor was located near the CT sensor in the sensor end-cap. A thin stainless steel diaphragm transmitted pressure from the outside water to the silicon diaphragm. The pressure sensor was an absolute pressure device with a time response in the 1 kHz range.

## 6. Perspective and Conclusions

Based on the analysis of published papers and open information from CTD manufactures, three research directions were proposed. Long time stability was an important requirement for CTD measurement. Interior calibration technologies of conductivity, temperature and depth were necessary and valuable. A negative temperature efficiency ceramic sensor had been used as internal standard of temperature sensor. Many inductive conductivity sensors had inner standards to maintain high stability. The second issue was the integration and minimization of the CTD system. Sensor chips could be further minimized. Many circuit components may be shared between each other. Compact package could decrease size, weight and power, avoid leakage and increase long-term stability. Integrated and minimized CTD sensors have more applications, such as flexibility and CTD sensors attached to ocean animals which can be a useful tool for the investigation of marine animals and marine environment monitoring [128,129,130,131,132]. The third direction involved the combination of artificial intelligence for hardware managements and data analysis. CTD should collect large amount of data with long-term stability and often communicate to researchers for data analysis. Smart and high-performance CTD sensors will be welcomed in ocean investigation. 

This review describes the latest developments of CTD sensors and the best properties of CTD sensors were displayed in detail. The principles, structure, materials and properties of many CTD sensors were demonstrated. Those frequently-used CTD products were tabled with performances. The working mechanisms of those CTD products were discussed. Three possible study directions of CTD sensors were proposed. This review will provide a wealth of information and cutting-edge data for researchers interested in the associated scientific societies.

## Figures and Tables

**Figure 1 sensors-23-00586-f001:**
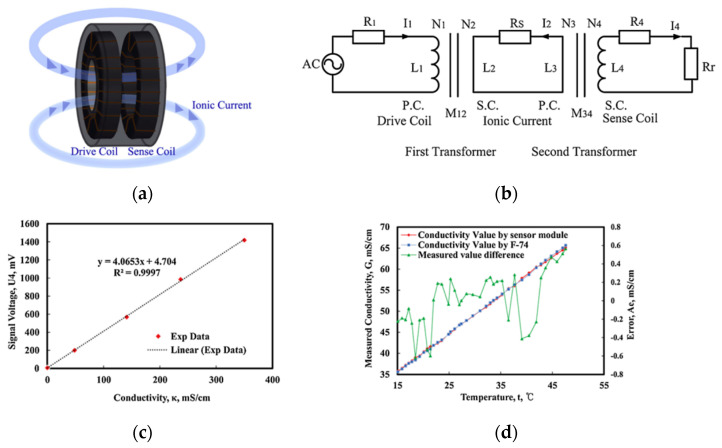
Inductive conductivity sensor: (**a**) schematic structure; (**b**) equivalent circuit; (**c**) experiment data with linear fitting result; (**d**) result of comparison test with the commercial water quality meter F-74.

**Figure 2 sensors-23-00586-f002:**
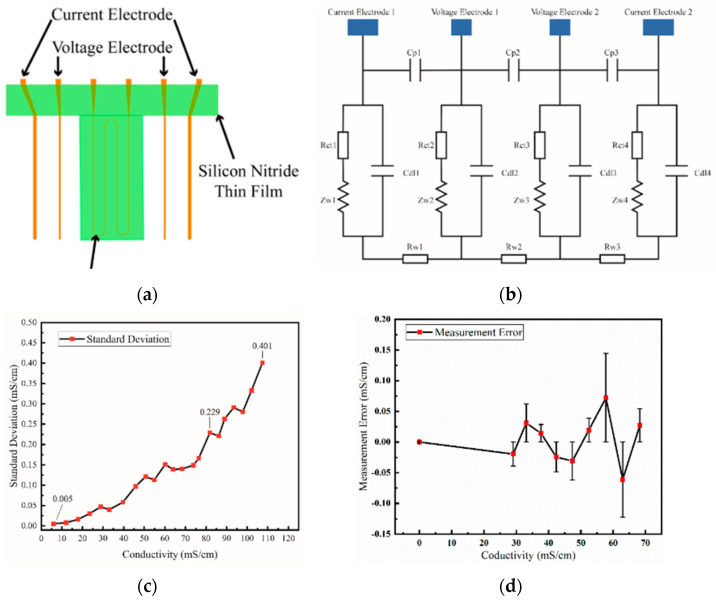
Four–electrode conductivity sensor: (**a**) chip structure; (**b**) equivalent circuit diagram; (**c**) result of measuring precising; (**d**) measurement error.

**Figure 3 sensors-23-00586-f003:**
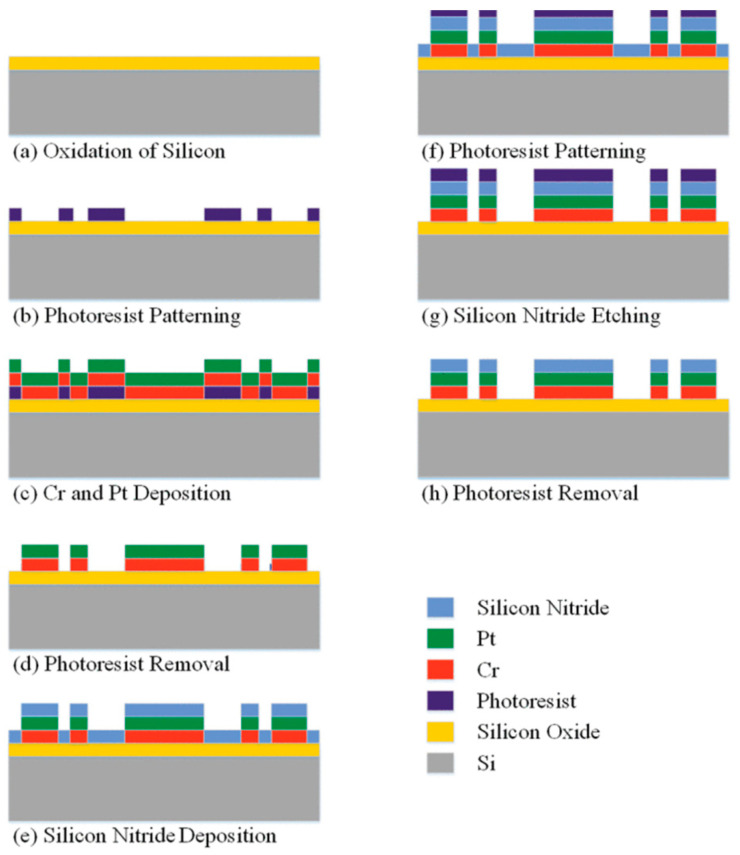
Batch fabrication process of the four-electrode conductivity sensor.

**Figure 4 sensors-23-00586-f004:**
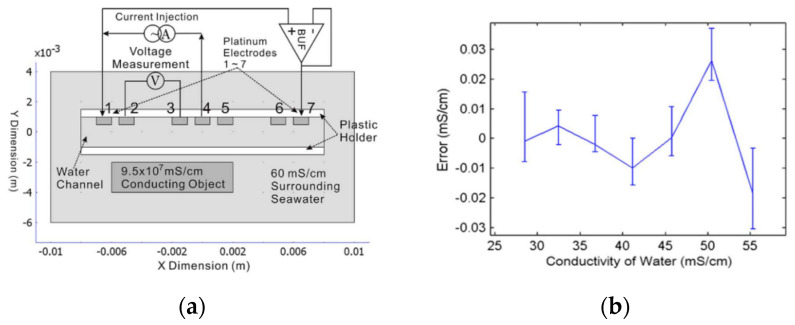
Seven–electrode conductivity sensor: (**a**) structure and measuring principle (**b**) errors in measuring seven conductivity points.

**Figure 5 sensors-23-00586-f005:**
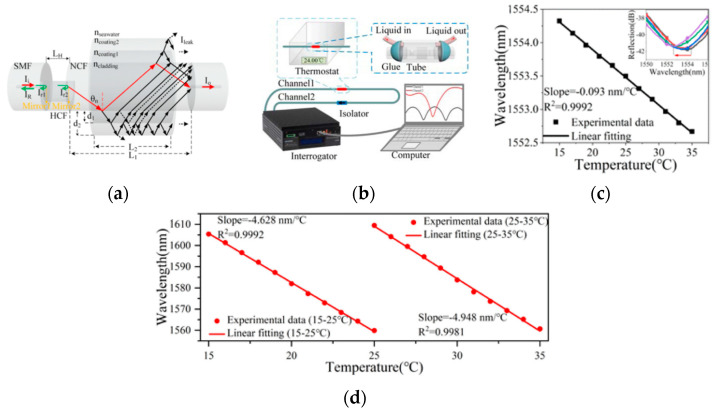
FPI (Fabry–Perot interferometer) optical fiber salinity sensor: (**a**) schematic diagram of sensing structure; (**b**) schematic diagram of experimental system; (**c**) reflecting spectrum to salinity (**d**) relationships between dip wavelength and temperature.

**Figure 6 sensors-23-00586-f006:**
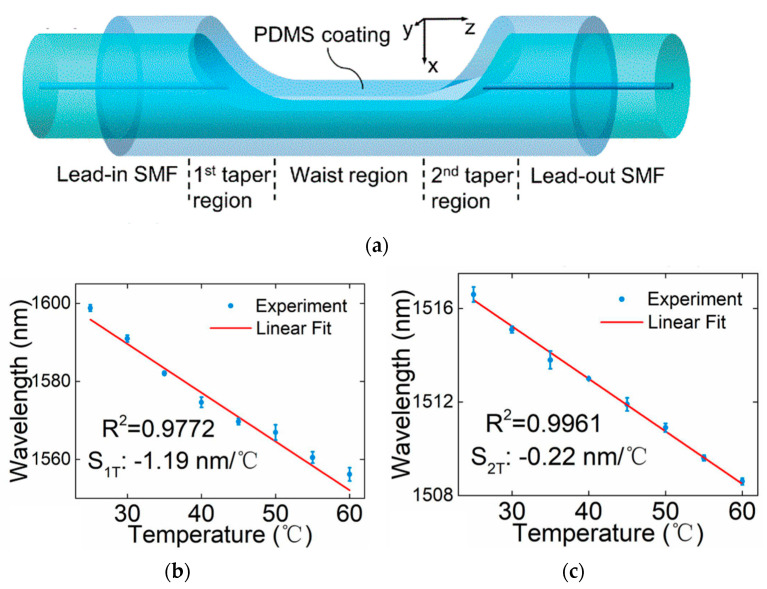
MZ type temperature sensor: (**a**) schematic diagram of coated microcavity; (**b**) relationships between wavelength dip on the transmission spectrum and temperature; (**c**) relationships between wavelength peak on the transmission spectrum and temperature.

**Figure 7 sensors-23-00586-f007:**
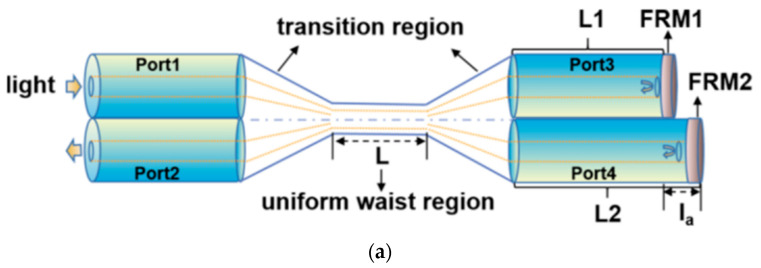
MFCL (microfiber couple loop) type temperature sensor: (**a**) schematic diagram of structure; (**b**) interference spectrum of different temperature; (**c**) relationships between dips of interference spectrum and temperature.

**Figure 8 sensors-23-00586-f008:**
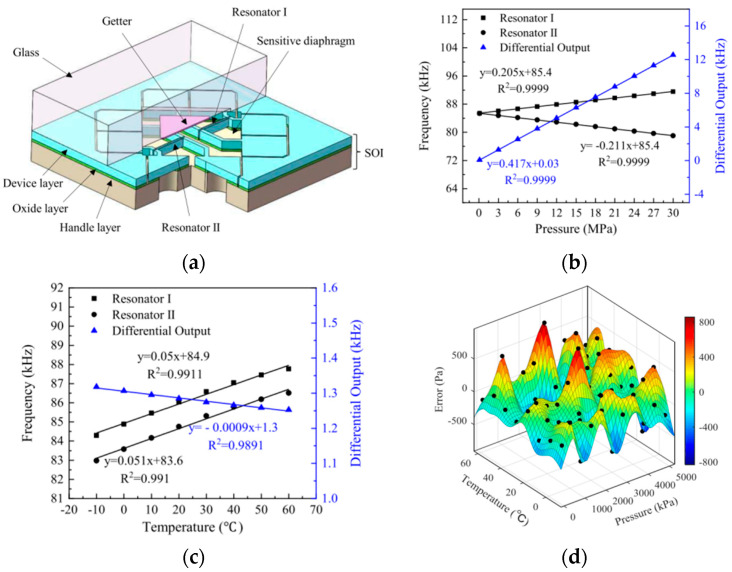
Resonant pressure sensor: (**a**) schematic diagram of structure; (**b**) high–pressure sensitivity in 20 °C; (**c**) temperature sensitivity under 110 kPa; (**d**) error of the pressure sensor after temperature compensations.

**Figure 9 sensors-23-00586-f009:**
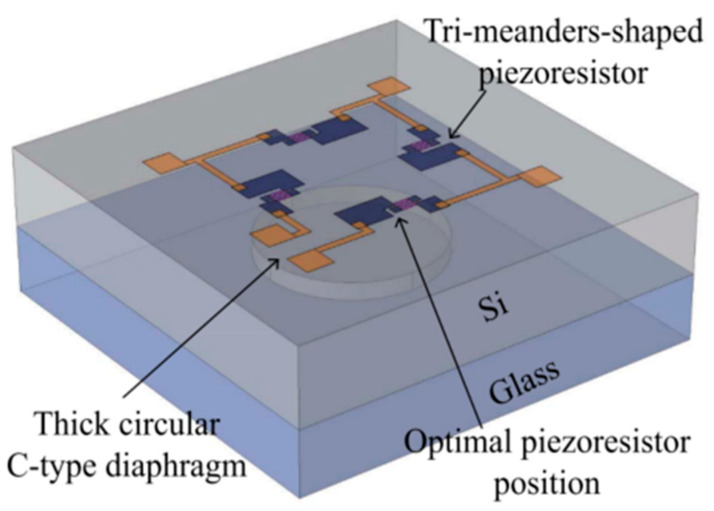
Structure of piezoresistive pressure sensor.

**Figure 10 sensors-23-00586-f010:**
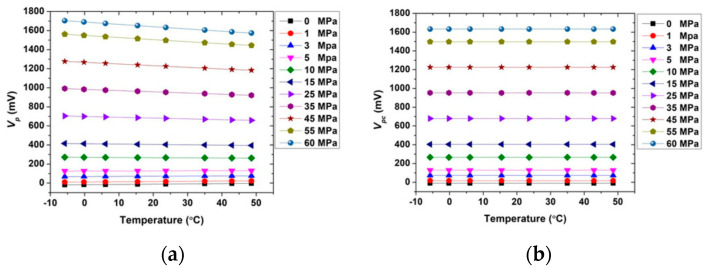
Temperature sensitivity change of pressure sensor (**a**) before compensation; (**b**) after compensation.

**Figure 11 sensors-23-00586-f011:**
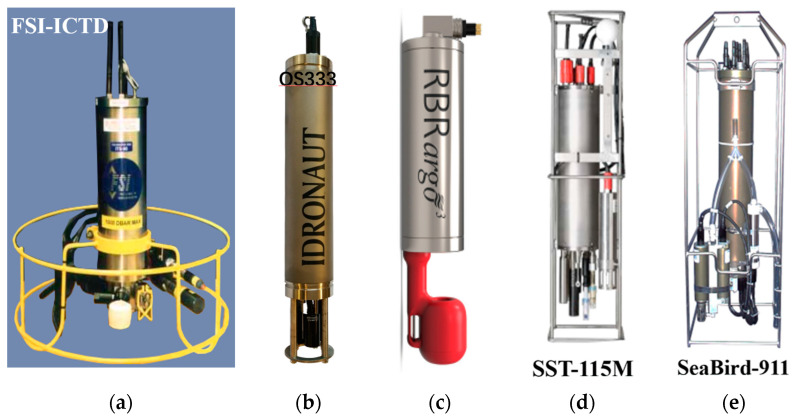
Representative CTD manufacturers and their products, (**a**) FSI-ICTD, (**b**) Idronaut-OS333, (**c**) RBR-argo3, (**d**) Sea Sun Tech-115M, (**e**) SeaBird 911plus.

**Table 1 sensors-23-00586-t001:** Relative error test result of FPI optical fiber salinity sensor.

Test	Temperature (°C)			Salinity (‰)		
	Actual Value	Measured	Relative Error	Actual Value	Measured	Relative Error
1	16	15.878	0.76%	5	4.726	5.48%
2	31	30.314	2.21%	20	19.228	3.86%
3	34	33.31	2.03%	20	21.049	5.25%
4	28	28.238	0.85%	30	29.209	2.64%
5	18	17.52	2.67%	40	39.218	1.96%

**Table 2 sensors-23-00586-t002:** Testing result of piezoresistive sensor under different pressure.

Pressure (MPa)	Output (mV)	Standard Deviation(mV)
0	−1.251	0.237
24	35.357	0.430
48	71.984	0.633
72	108.634	0.839
96	145.297	1.043
120	181.971	1.242

**Table 3 sensors-23-00586-t003:** Average values of piezoresistive sensors parameters.

Parameter	Average Value	Standard Deviation
Resistance (kΩ)	3.591	0.039
Sensitivity (mV/V/MPa)	0.425	0.00263
Nonlinearity (%FS)	0.0156	0.00204
Repeatability (%FS)	0.00685	0.00163
Hysteresis (%FS)	0.00624	0.00190
Accuracy (%FS)	0.0182	0.00208

**Table 4 sensors-23-00586-t004:** Summary of representative commercially-available CTD products.

Company/Type/ Reference		Conductivity (mS/cm)			Temperature (°C)			Depth (Bars)			Response Time (s)	Max Sampling Frequency (Hz)
		Range	Accuracy	Resolution	Range	Accuracy	Resolution	Range	Accuracy	Resolution		
1	Aanderaa/SeaGuard [113]	0–75	±0.018	0.002	−4–36	0.02	0.001	0–600	±0.02%FS	<0.0001%FS	C: 3, T&D: 10	1.0
2	AML/AML-3 XC [114]	0–90	0.006	0.001	−5–45	0.01	0.001	0–600	0.1%FS	0.02%FS	C: 0.025, T: 0.1, D: 0.01	20.0
3	FSI/ICTD [115]	0–70	±0.002	0.0001	−2–35	0.002	0.00005	0–700	±0.01%FS	0.0004%FS	C: 0.001, T: 0.15, D: 0.025	32
4	HACH/HydroCAT [116]	0–70	±0.003	0.0001	−5–45	±0.002	0.0001	0–35	±0.1% FS	0.002%FS		0.17
5	Idronaut/OS333 [117]	0–90	0.0015	0.0001	−5–50	0.0015	0.0001	0–700	0.05% FS	0.0015% FS	C&T&D: 0.05	28
6	JFE/RINKO-Profiler [118]	0.5–70	±0.01	0.001	−3–45	±0.01	0.001	1–100	±0.3% FS	0.003	C&T: 0.2, D: 0.1	10
7	NBOSI/Cabled CT [119]	0–60	±0.01	±0.0001	0–30	±0.005	±0.0001	NA	NA	NA	C&T&D: 0.4	5
8	Ocean Sensor Systems/OSSI010003B [120]	NA	NA	NA	−10–65	±1.25	0.0625	0–10	±0.05% FS	0.0033	NA	30
9	Ocean Sensors/OS200 CTD [121]	0.5–65	0.02	0.001	0–30	0.01	0.001	0–100	0.50%FS	0.005%FS	C&T&D: 0.02	0.3
10	RBR/argo^3^CTD [122]	0–85	±0.003	0.001	−5–35	±0.002	0.00005	0–600	±0.05% FS	0.005%FS	T: 0.7, D: 0.01	8
11	Sea Sun Technology/CTD 115M [123]	0–70	±0.002	0.0005	−2–36	±0.002	0.0005	0–50	0.05% FS	0.002%FS	C&T&D: 0.15	5
12	SEABIRD/SBE 911plus CTD [124]	0–70	±0.003	0.0004	−5–35	±0.001	0.0002	0–1050	±0.015%FS	0.001%FS	C&T: 0.065, D: 0.015	24
13	SonTek/castaway CTD [125]	0–100	±0.005	0.001	−5–45	±0.05	0.01	0–100	±0.25%FS	0.001	T: 0.2	5
14	WTW/CastAway-CTD [126]	0–100	±0.005	0.001	−5–45	±0.05	0.01	0–19	±0.25%FS	0.001	NA	NA
15	YSI/ProDSS [127]	0–200	±0.5% of reading	0.001	−5–50	±0.15	0.1	0–10	±0.0004	0.0001	NA	NA

## Data Availability

No new data were created or analyzed in this study. Data sharing is not applicable to this article.

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
