# Peer review of "CTD Sensors for Ocean Investigation Including State of Art and Commercially Available"

_sensors, 2023, doi:10.3390/s23020586_

Round 1

Reviewer 1 Report

The manuscript reviews the sensors of Temperature Conductivity Depth for ocean investigation. The manuscript's topic is attractive and meaningful, and the logical outline is clear, but making the following corrections will strengthen the article. 

1) The title could cover extensive content on sensor research, development, and application. However, this article mainly focuses on research, and commercial products, which is not enough to fit the title. Therefore, the authors need to reconsider the title.

2) It is suggested to add recent studies of quartz resonate tuning forks, vibration diaphragm quartz resonate, and strain gauge pressure sensors. Because it is illustrated in the commercial part, for instance, the Digiquartz sensor applied in SBE 911 plus and the strain gauge pressure sensor used in OS 200.

3) It is necessary to rigorously evaluate the referenced research in each chapter rather than research work stack or put together.

4) It is suggested to compare the performance of sensors based on various principles and to analyze their advantages and disadvantages.

5) The static and dynamic response properties are essential evaluation items for the sensing device and instrument. Therefore, adding the response time and sampling frequency in the commercial part is suggested.

6) Temperature compensation is necessary to promote the accuracy of the depth sensor. Therefore, it is suggested to review the compensation principle and studies.

7) Brief comment describing each symbol is necessary—for example, formula (2) in chapter 4, Depth Sensors.

8) Figure resolution is low—for example, Figure 3, Figure 6a, etc.

9) The abstract should be firm and focus on key points rather than a stack of numbers.

Reviewer 2 Report

Title of this manuscript is “A review on sensors of Conductivity Temperature Depth for Ocean Investigation”, and catalog values of a CTD sensor provided by the manufacturer are listed in the abstract. However, sensors that are under development but not yet in practical use are presented in large part of this manuscript (sections 2, 3, 4). Moreover, specifications of the sensors under development are not compared with the commercially available CTD sensors (section 5). Therefore, the purpose of this manuscript is not clear. Judging from the content of this manuscript, I think the title should be changed to such as a review on sensors of salinity, temperature, and depth under development for ocean investigation, and the characteristic values of those sensors should be compared with those of commercially available sensors.

Overall, I think the authors should reexamine the references and cite more appropriate references. For example, for lines 32-33, almost the same sentence is stated in the reference [1], but no evidence is provided that it is less than 5% in the reference [1]. Similarly, in the introduction, as references for the fact that ship-based observations provide more accurate information than satellite observations or underwater observations, the manuscript cites three papers [3-5] that specialize in microstructure observation using fast response thermometers. However, there should be more appropriate references showing that ship-based observations are highly accurate. In ship-based CTD observations, there are fewer restrictions on sensor size, power supply capacity, memory capacity, etc., and in addition to the ability to perform pre- and post-calibration of sensors, the availability of in-situ water sampling data for sensor correction is thought to contribute significantly to high accuracy.

In this manuscript, refractive index sensors for salinity measurement are treated as “optical conductivity” sensors. However, in principle, the refractive index of seawater is related to the density of seawater (the Lorentz-Lorenz equation, see Le Menn and Nair, 2022). Therefore, Refractive index-Temperature-Depth sensor can estimate “Absolute Salinity”. Meanwhile, Conductivity-Temperature-Depth sensor can estimate “Practical Salinity”. The difference between the two should be clear throughout the text.

If temperature and pressure are measured (or these two are constant) in addition to the refractive index measurement, Absolute Salinity can be estimated, so it acts as an Absolute Salinity sensor. If Absolute Salinity and pressure are measured (or these two are constant) in addition to the refractive index measurement, temperature can be estimated, so it acts as a temperature sensor. If temperature and Absolute Salinity are measured (or these two are constant) in addition to the refractive index measurement, so it acts as a pressure sensor. It is necessary to give some background on these for the reader.

Absolute Salinity estimation based on refractive index measurements is reviewed in detail by Le Menn and Nair (2022) and should be cited.

Le Menn and Nair (2022): Review of acoustical and optical techniques to measure absolute salinity of seawater. Frontiers in Marine Science, https://doi.org/10.3389/fmars.2022.1031824

There are too many errors throughout the text to make reviewing difficult. For example, Lines 145-146 states “The fabrication of the sensor is explained in Figure 2(c).”, but the figure caption states “(c) Result of measuring precising(?)”. For another example, Lines 323-324 states “The pressure sensitivity of the sensor is 3.96nm/MPa which is tested in a range of 0MPa to 0.48MPa, shown in Figure 9.”, but the figure caption states “Structure of piezoresistive pressure sensor”.

Followings are comments I noticed on a quick read.

Lines 18-20: CTD means Conductivity-Temperature-Depth. Therefore, optical, acoustic wave and radio wave sensors are not included in category of CTD. In addition, radio wave sensors are NOT described in the main body.

Lines 20-24: It should be made clear that accuracies are typical values claimed by the manufacturer.

Line 72: The temperature sensor of SBE 911 plus is not platinum resistance thermometer but NTC thermistor.

Line 83: seawater47 -> seawater [47]

Lines 87-88: Conductivity is NOT salinity, so the line should be “ranges from ** to ** S/m”.

Line 129: Range of Practical Salinity is defined only 2-42 PSU. How do you estimate the error far outside of the definition (100-500 PSU).

Section 3: Thermistors are also commonly used (e.g. SBE3plus temperature sensor).

Section 4: Resonant silicon crystal sensors, which are very cheaper than resonant quartz crystal sensors, are also commercially available (e.g. GE Druck and Yokogawa Electric Co.).

Lines 276, 299, 321: 3.1 -> 4.1, 3.2 -> 4.2, 3.3 -> 4.3

Line 329: dip1, dip1 -> dip1, dip2

Line 330: Table 1 is “Compare FPI optical fiber sensor with thermometer and Abbe refractometer”. Table 2 is also probably different from the subject of this description.

Table 2: The table should be separated on the left side and the right side because it is confusing.

Lines 355-358: It should be made clear that these values are typical values claimed by the manufacturer. As each manufacturer may have different definitions of “accuracy”, it may not be appropriate to directly compare the claimed values from different manufacturers.

Line 358: 100500 m -> 10500 m

Line 414: This information is old. SBS should have been using one NIST traceable TPW cell for 10 years. Please contact the manufacturer.

Section 5: There is too much difference in the amount of information for each manufacturer. For example, SBS has a one-page description, but JFE Advantech has no description at all. As described at lines 358-360, several other properties, important information such as stability, response time, sampling, power, housing material, size, and weight should be compared.

Round 2

Reviewer 1 Report

It is well organized and sincerely revised. It is publishable.

Reviewer 2 Report

Throughout the text, sensors based on refractive index measurements are still referred to as CTD sensors. CTD is an abbreviation for Conductivity-Temperature-Depth, so optical salinity (refractive index) sensors are not included in CTD. I recommend using STD (Salinity-Temperature-Depth) or more explicitly Refractive index-Temperature-Depth sensors for such optical sensors (e.g. Line 20, lines 79-83, Line 129, and Section 6).

And there are still many errors (listed below). Please revise carefully.

Lines 20-21: The “sensors based on acoustic wave and radio wave” should be described not only in Abstract but also in the main text with references. As with refractive index sensors, it is possible to measure Absolute Salinity from measurements of sound velocity, temperature, and pressure, but such a sensor is not called a CTD. STD (Salinity-Temperature-Depth) or more explicitly Sound velocity-Temperature-Depth sensor should be used.

Line 92: Absolute salinity -> Absolute Salinity (SA)

Lines 119-120, 129: optical conductivity sensors -> optical salinity sensors

Line 121: conductivity -> salinity

Line 147: Figure 1(e) -> Figure 1(d)

Line 148: glide -> glider

Line 151: the error expanded -> the error reduced? (Since the magnitude of -0.178 is smaller than the magnitude of -0.225)

Line 158: MEMS sensor -> MEMS

Lines 184-185: There is no table corresponding to this description.

Line 190: wo -> to?

Lines 223 and 227: conductivity sensor -> salinity sensor

Lines 252, 256, 257: There is no figure corresponding to this description.

Lines 260, 271, 280, 318: Please indicate what “SMF, MAA, TEAB, FRM, SOI” stands for.

Line 287: There is no table corresponding to this description.

Line 296: Please indicate what “h, P, g, rho” stands for.

Line 307: 0.1FS -> 0.1%FS

Line 314: noew -> now

Line 360: m/MPa -> mV/MPa?

Line 361: 0.43 -> 0.43%?

Table 2: Standard Deviation -> Standard Deviation (mV)

Line 369: Table 3 is not referred in the text.

Line 378: in shown in -> shown in

Line 378: There is no figure (Figure 7(a) and Figure 7(b)) corresponding to this description.

Line 381: There is no table (table 1) corresponding to this description.

Line 384: 257.79nm -> 257.79nm/MPa?

Line 492: Wie -> Wien
